Does Drosophila sechellia escape parasitoid attack by feeding on a toxic resource?

Salazar-Jaramillo Laura lauraalazar@gmail.com 1 2
Wertheim Bregje 2
1 Vidarium-Nutrition, Health and Wellness Research Center , Medellin , Colombia
2 Groningen Institute for Evolutionary Life Sciences (GELIFES), University of Groningen , Groningen , Netherlands
Ramirez Claudio
Electronic publication date: 2021 Jan 6
Publication date: 2021
Volume: 9
Electronic Location ID: e10528
Received 2020 Apr 3; Accepted 2020 Nov 18
Copyright: ©2021 Salazar-Jaramillo and Wertheim
Copyright year: 2021
Copyright holder: Salazar-Jaramillo and Wertheim
License: This is an open access article distributed under the terms of the Creative Commons Attribution License, which permits unrestricted use, distribution, reproduction and adaptation in any medium and for any purpose provided that it is properly attributed. For attribution, the original author(s), title, publication source (PeerJ) and either DOI or URL of the article must be cited.
License URL: https://creativecommons.org/licenses/by/4.0/

Keywords: Drosophila sechellia, Parasitoid wasps, Host shift, Trait loss, Noni

Funding: Royal Netherlands Academy of Arts and Sciences Reference UPS/297/Eco/1413 The University of Groningen This work was supported by the Royal Netherlands Academy of Arts and Sciences (Reference UPS/297/Eco/1413) and The University of Groningen. The funders had no role in study design, data collection and analysis, decision to publish, or preparation of the manuscript.

==============================
Host shifts can drastically change the selective pressures that animals experience from their environment. Drosophila sechellia is a species restricted to the Seychelles islands, where it specializes on the fruit Morinda citrifolia (noni). This fruit is known to be toxic to closely related Drosophila species, including D. melanogaster and D. simulans, releasing D. sechellia from interspecific competition when breeding on this substrate. Previously, we showed that larvae of D. sechellia are unable to mount an effective immunological response against wasp attack, while larvae of closely-related species can defend themselves from parasitoid attack by melanotic encapsulation. We hypothesized that this inability constitutes a trait loss due to a reduced risk of parasitoid attack in noni. Here we present a lab experiment and field survey aimed to test the hypothesis that specialization on noni has released D. sechellia from the antagonistic interaction with its larval parasitoids. Our results from the lab experiment suggest that noni may be harmful to parasitoid wasps. Our results from the field survey indicate that D. sechellia was found in ripe noni, whereas another Drosophila species, D. malerkotliana, was present in unripe and overripe stages. Parasitic wasps of the species Leptopilina boulardi emerged from overripe noni, where D. malerkotliana was the most abundant host, but not from ripe noni. These results indicate that the specialization of D. sechellia on noni has indeed drastically altered its ecological interactions, leading to a relaxation in the selection pressure to maintain parasitoid resistance.

Introduction

Host shifts are considered to be of major importance in the ecology and evolution of organisms (Nyman, 2010). Adaptation to feeding on novel host-plant species is largely believed to promote speciation and to be a key factor underlying the evolutionary diversification of insects (Ehrlich & Raven, 1964; Matsubayashi, Ohshima & Nosil, 2009; Futuyma & Agrawal, 2009; Nyman, 2010). Among the factors that explain speciation after a host shift are access to new food sources, changes in the competitive dynamics among species and enemy-free spaces (Feder, 1995; Janz, Nylin & Wahlberg, 2006; Weingartner, Wahlberg & Nylin, 2006; Nyman, Bokma & Kopelke, 2007; Hardy & Otto, 2014). Host shifts often appear to be linked to evolutionary changes in life-history traits. Examples include how recognition and avoidance of specific food affects mating in pea aphids (Caillaud & Via, 2006), the evolved mechanisms to cope with toxic chemical compounds in cactophilic fruit flies (Matzkin, 2012), and the shifting of the life cycle to coincide with host availability in apple maggots (Dambroski & Feder, 2007). Whole-genome sequencing is providing new insights into the causes or consequences of host shifts from the associated changes in the genome. For example, in giant and red pandas, from two distinct taxonomic families, convergent evolution was found in the genetic regulation of the pseudo-thumb that is associated with bamboo eating in both species (Hu et al., 2017). Moreover, in phytophagous beetles expansion of gene families involved in detoxification of plant secondary compounds has been described (Seppey et al., 2019).

The sequencing of several Drosophila species, including the dietary specialist D. sechellia also made a great contribution to the study of molecular and genomic consequences of host shift (McBride, 2006; Clark et al., 2007; Whiteman & Pierce, 2008). D. sechellia is restricted to the Seychelles islands in the Indian Ocean, where it specialized on the fruit Morinda citrifolia (commonly known as noni) (Louis & David, 1986; Gerlach, 2009). The noni is toxic to most Drosophila species (Farine et al., 1996). A study on the biochemical basis of the toxicity of noni revealed that D. sechellia was five to six times more resistant than D. melanogaster to one of the toxic compounds, octanoic acid (Legal, Chappe & Jallon, 1994). Genomic changes associated with D. sechellias specialization on noni have been described for smell and taste receptors, detoxification genes (McBride, 2006; Matsuo et al., 2007) and genes associated with its resistance to octanoic acid (Jones, 1998; Andrade-López et al., 2017; Lanno et al., 2019). Octanoic acid is present at high concentration in the ripe stage of the fruit, but less so in the overripe rotten and unripe stages (Legal, Chappe & Jallon, 1994). In the Seychelles, D. sechellia is found abundantly and preferentially on M. citrifolia fruits, with a small proportion of adults also found in other fruits (Matute & Ayroles, 2014). Adults of D. simulans have also been reported in M. citrifolia but it is not clear whether they are able to breed in this fruit (Matute & Ayroles, 2014). It is believed that resistance of D. sechellia to the octanoic acid levels during the highest peak in toxicity provides this species with a unique ecological niche among coexisting and potentially competing Drosophila species (Andrade-López et al., 2017), thus minimizing competition.

While studying the evolution of the immune response to parasitoid attack in Drosophila species, we found phenotypic and molecular evidence for a potential consequence of D. sechellia host shift to noni, namely the loss of the ability to mount an immune response against parasitoid wasps. Parasitoids are insects that lay their eggs in or on other insects, and the developing parasitoid larvae eat and kill their host (Godfray, 1994). Parasitoids are common and can constitute a significant mortality factor for Drosophila species (Janssen et al., 1987; Wertheim et al., 2006; Fleury et al., 2009). Various defensive mechanisms evolved in Drosophila to survive parasitoid infection, but species and populations vary largely in defensive ability, including the complete absence of parasitoid resistance in several species (Eslin & Doury, 2006; Havard et al., 2009). Within the phylogeny of Drosophila, at least 3 different types of specialized hemocytes (i.e., insect blood cells) evolved that are involved in immune responses to sequester and kill parasitoid eggs. Each of these specialized hemocytes is restricted to sublineages of the phylogeny (Havard et al., 2012; Salazar-Jaramillo et al., 2014; Márkus et al., 2015)), implying parasitoid resistance evolved repeatedly and is a derived trait for subsets of species. Besides immunological mechanisms, behavioral responses can have an important role, for example in the choice of diets for prophylaxis, therapy or compensation against pathogens, particularly in herbivorous insects (Singer, Mason & Smilanich, 2014). In D. melanogaster it has been shown that exposure to ethanol in the fly medium may reduce wasp oviposition, and that this protection is more effective against a generalist wasp than a wasp that specializes on D. melanogaster (Milan, Kacsoh & Schlenke, 2012). It has been repeatedly reported that D. sechellia is unable to defend itself against the infection of parasitoid wasps of the genera Asobara (Eslin & Prévost, 1998; Salazar-Jaramillo et al., 2014) and Leptopilina (Schlenke et al., 2007), as was also reported for several other Drosophila species (e.g., D. subobscura, D. persimilis, D. virilis (Havard et al., 2012)). Lack of immunity may be because defenses did not evolve, were bested by parasitoid counter-adaptations, or because an evolved trait was lost. In the case of D. sechellia, we found evidence it was a secondary loss of an immune response, and we query whether this is a consequence of its host shift to noni.

The immunological defense that is used by closely related species of D. sechellia is termed melanotic encapsulation and relies on the differentiation of lamellocytes. The mechanism of melanotic encapsulation starts with the detection of the parasitoid egg as foreign body, the egg is then surrounded by multiple layers of hemocytes (blood cells) and finally fully melanized. This kills the parasitoid egg, and enables the host to survive the parasitoid attack (Lemaitre & Hoffman, 2007). The differentiation and mobilization of hemocytes is a critical step in this process (Fauverque & Williams, 2011). In D. melanogaster three types of differentiated blood cells have been described: (1) plasmatocytes, which perform phagocytosis of bacteria and other small pathogens and are also recruited in the cellular capsules around parasitoid eggs, (2) crystal cells, which store the precursors of the melanin that is deposited on invading pathogens (Pech & Strand, 1996; Williams, 2007) and (3) lamellocytes, which are large, adhesive and flat cells that form the cellular layers around the foreign bodies (e.g., parasitoid eggs) and contain precursors for melanization. Previously we showed that the production of lamellocytes is restricted to a sublineage of the Drosophila phylogeny inside the melanogaster subgroup (Salazar-Jaramillo et al., 2014). While D. sechellia falls within this sublineage and can produce lamellocytes in response to parasitoid attack, it only does so in very low concentrations, and its immune defense against parasitoids is largely ineffective (Eslin & Prévost, 1998; Salazar-Jaramillo et al., 2014). Moreover, its transcriptional response to parasitoid attack is compromised, and several genes that are up-regulated in the other species in response to parasitoid attack show substantial sequence divergence or contained loss-of-function mutations in D. sechellia (Salazar-Jaramillo et al., 2014; Salazar-Jaramillo et al., 2017).

In two comparative studies, one on genomes and the second on transcriptomes, we revealed molecular signatures associated with a loss of resistance. In the comparative genomic study of 12 Drosophila genomes, we showed large sequence changes in several of the putative immunity genes uniquely in the genome of D. sechellia. In particular, two immune genes showed a potential loss of function sequence variation only in D. sechellia: (1) Tep1, which facilitates the recognition of pathogens, activation of immune pathways and phagocytosis (Dostálová et al., 2017) and (2) PPO3, which is expressed in lamellocytes and contributes to melanization during the encapsulation process (Dudzic et al., 2015). In Tep1, we found missing exons; in PPO3, we found that the dN/dS ratio was close to unity, suggesting neutral evolution. The disproportionate rate of nucleotide substitution was later shown by another study to correspond to an inactivating mutation due to the introduction of a stop codon (Dudzic et al., 2015). When we examined the expression of both genes through qPCR, they were not up-regulated in D. sechellia in response to parasitoid attack while they were strongly up-regulated in both D. melanogaster and D. simulans (Salazar-Jaramillo et al., 2014).

In a comparative transcriptomic study, we further revealed evidence of the failure of a functional immune response in D. sechellia to parasitoid attack (Salazar-Jaramillo et al., 2017). Although D. sechellia showed upregulation of a few immune genes that react to a general immune challenge (e.g., the homologs of Mtk, DptB, PGRP-SB1), it failed to upregulate most of the genes that were upregulated during the cellular immune response against parasitoids in the closely related species D. melanogaster and D. simulans. These genes included TepI, PPO3, CG4259, CG4793, TotA and Spn88EB. Based on these combined results we proposed the hypothesis that D. sechellia possesses the genes for melanotic encapsulation, but that they no longer function properly in response to parasitoid attack (Salazar-Jaramillo et al., 2017).

We hypothesize that the loss of immunological defenses against parasitoid wasps may be caused by the host shift of D. sechellia to the toxic noni fruits. The toxicity of the noni could either affect directly the parasitoid wasps, for example preventing oviposition, or indirectly via the fly larvae where the wasps develop. In both cases, if noni are toxic to the local parasitoid population, D. sechellia’s shift to this breeding environment may protect it from attack by parasitoid wasps, releasing it from its natural enemy, which could result in the degeneration of the immune defenses (Salazar-Jaramillo et al., 2014). Fundamental to this hypothesis is the effect of noni on the development of parasitoid wasps, and its emergence from D. sechellia that feed and breed on noni. To investigate whether breeding on noni indeed provides an enemy-free space, we here present the results of a lab experiment and a field survey in Cousin, one of the Seychelles Islands. The two questions we addressed were: (1) Is noni harmful to parasitoid wasps? (2) Do parasitoid wasps infect D. sechellia when they are feeding and breeding on noni? To answer the first question, we set up a lab experiment with the wasp Asobara citri on hosts fed with a commercial noni extract. To answer the second question, we did a field survey to test whether D. sechellia flies that breed and feed on noni were infected by parasitoid wasps. Cousin has the status of Special Reserve for nature conservation and due to its conservation status, it is not allowed to extract live material, which limited our sample sizes considerably, and prevented us from bringing local specimens to the lab for additional testing. Both the lab experiment and the field study may not be conclusive on their own, but they provide compelling support for our hypothesis in the light of the multiple lines of evidence.

Materials and Methods

Our aim was to study for the first time the tritrophic system of parasitoid wasps emerging from hosts that feed and breed on noni. For this purpose, we carried out a lab experiment and a field survey. The lab experiment focused on assessing the survival and emergence of the parasitoid wasp Asobara citri exposed to noni media in the two common host species, D. melanogaster and D. simulans. The field study sought to observe the drosophilids and wasps that emerge from different stages of noni. We collected noni fruits at different levels of maturity, and reared out insects that developed on these fruits.

Lab experiment

In order to test the effect of noni on parasitoid wasps, we used the host species D. melanogaster and D. simulans and the wasp species Asobara citri. D. sechellia could not be used in the experiment as this species lays significantly less eggs than the other two species (Jones, 2004), which would hamper a proper comparison (unfortunately even in the commercial noni extract). The fly species are the genome project strains from the Drosophila Stock Center (UC San Diego University) (Drosophila 12 Genomics Consortium 2007). The parasitoid strain of A. citri was collected in Ivory Coast, West Africa, in 1995, and has been maintained on D. melanogaster at 25 °C. For the survival and parasitization assays, adult flies were placed on petri dishes of 70 mm diameter filled with standard medium (26 g dried yeast, 54 g sugar, 17 g agar and 13 mL nipagine solution per litre) and a layer of either noni extract (‘Hawaiian Health Ohana’ brand) or yeast. They were incubated overnight for egg-laying, after which the adult flies were removed from the petri dishes. Eggs were transferred to new petri dishes at standardized density (25 eggs/petri dish), and kept at 25 °C. After two days, when the larvae were in the second-instar stage, a female of A. citri was introduced to half of the petri dishes for 24 hours, while the rest of cultures were used as a non parasitized control condition (n = 4 petri dishes per fly species (D. melanogaster, D. simulans), per medium type (noni, yeast) and per parasitization treatment (parasitoid, control). After pupation, the fly pupae were transferred to vials to complete development away from the potential toxic compounds. This would reflect the natural behaviour of Drosophila fruit flies, which often pupate outside the oviposition substrate (Sameoto & Miller, 2012). The number of adult flies and wasps that emerged from the petri dishes was scored. The adult flies were collected and squashed between two glass slides under a stereo-microscope to assess whether they had been parasitized and had encapsulated the parasitoid egg, following similar protocols as described in Salazar-Jaramillo et al. (2014). Generalized Linear Models (glm) fitted to a quasibinomial distribution to take into account overdispersion were implemented in R version 3.6.3 to analyse: (1) survival of unparasitized controls as the number of adult flies of D. simulans and D. melanogaster that developed in each of two media, yeast and noni extract; (2) success of parasitization as the emergence of wasps in the two types of hosts media and (3) parasitization rate, as the number of parasitized hosts in the two types of media (Table 1). (1) GLMsuccess|failure∼host_species+medium,family=quasibinomial

Since we used a commercial extract of the noni fruit and we could only test the response of D. melanogaster and D. simulans against a (West-African) parasitoid wasp not known to occur in the Seychelles we next investigated the occurrence of parasitic wasps in D. sechellia feeding and breeding on noni in the wild.

Table 1 Variables used as success and failure in the GLM model to analyse the corresponding query.

Each query for the analysis of the lab experiment is statistically tested using a GLM model (1) and the corresponding variables listed as success and failure (survival: total number of emerged adult insects, mortality: initial larvae - emerged adult insects, wasps: number of emerged wasps, capsule_flies: number of flies with a capsule, unparasitized_flies: number of adult flies without a capsule).

Query	Success	Failure	
survival of unparasitized hosts	survival	mortality	
success of parasitization	wasps	unparasitized _flies + flies _capsule	
parasitization rate	wasps + flies _capsule	vs unparasitized _flies	

Field survey

We did a survey of drosophilid flies and parasitoid wasps occurring on wild noni fruit on Cousin island in the Seychelles (Permit approved by Seychelles Bureau of Standards Ref A0157). Cousin Island is a nature reserve characterized by the presence of indigenous and endemic forest (mixed Pisonia, Noni and Ochrosia). We collected noni fruit that had fallen off the plant, at different stages of maturity, (corresponding to the approximate stages 2–4 described in Chan-Blanco et al. (2006)). In total 28 noni fruit were collected, ranging from unripe (green coloration and high hardness, “stage 2”), to ripe (yellow and high hardness, “stage 3”), to overripe (light yellow or white and moderate hardness, stage “stage 4”). We also placed a small number (12 samples) of alternative fruit substrates out as bait (papaya, banana), to explore whether other drosophilid and parasitoid species were part of the community that did not occur on noni. Both the noni and alternative fruits were placed in plastic containers to capture larvae that would leave the fruit for pupation. The containers and fruits were left open at the site of collection to enable further oviposition by insects. Thereafter, the containers were closed with a piece of gauze to ensure that all insects that emerged from the fruit could be retrieved. The containers were brought to the field station and checked regularly for any emerging insects. Of the 28 samples, 17 samples had at least one drosophilid species. Emerged adult insects were collected and preserved in 70% ethanol for taxonomic identification.

Species identification

Drosophilidae species could be identified to the species level by Prof Marie-Louise Cariou (Laboratory of Evolution Genomes and Speciation, CNRS-Paris) using morphological traits (genitalia and palp morphology) (Lachaise et al., 2008). Emerged wasps were identified through morphological characters (antennae, wing venation, scutellum and shape of the scutellar cup) and through sequencing of a Cytochrome Oxidase I fragment (Lue et al., 2016).

Tables of raw data are shown in the Supplementary Material and, together with the analysis, deposited in the repository: https://github.com/lauraalazar/Dsechellia-parasitoid).

Results

Lab experiment

We found that D. melanogaster and D. simulans showed significant differences in survival, but this was not related to the exposure to the commercial noni extract (Fig. 1A). The survival of D. simulans was significantly higher than the survival of D. melanogaster both in the control samples (where no parasitoid was introduced) (glm, F = 15.65, DF = 14.24, P = 0.00364) and in the parasitization samples (glm, F = 7.4, DF = 14, P = 0.021). While the survival of the fly hosts was not significantly affected by the exposure to noni, the medium had a significant effect on the survival and parasitization of parasitoid wasps (Fig. 1B). The proportion of wasps that emerged from the petri dishes (i.e., success of parasitization) was significantly lower on noni medium than on yeast medium (glm, F = 12.64, DF = 13, P = 0.004). This reflected primarily fewer hosts that were attacked by parasitoids (i.e., parasitization rate) on noni than on yeast (glm, F = 13.14, DF = 13, P = 0.00348). A striking observation was a high mortality on noni medium among the parasitoids that were used for the parasitization treatment. When wasps were removed after 24 hours of parasitization, over half of the individuals parasitizing on noni extract were found dead, whereas none of the individuals on yeast had died. This will at least have contributed partially to the higher number of larvae that escaped parasitism. These results indicate that prolonged exposure to the noni extract is likely to be lethal for the Asobara wasp, while it surprisingly did not affect the survival for the Drosophila (neither adults nor larvae).

Figure 1 Lab experiment on the survival and parasitization of hosts on noni extracts.

The numbers of emerged adult insects were compared for (A) the unparasitized host species D. melanogaster and D. simulans and (B) the hosts that were exposed to the parasitoid wasp A. citri. Hosts that were parasitized yielded a wasp (i.e., successful parasitization) or a fly with an encapsulated parasitoid egg (i.e., unsuccessful parasitization); hosts that died during development may or may not have been parasitized. The differences between host species and medium type in (A) host survival and (B) host parasitization success and rate were analysed with glms (see methods); significance is indicated as: “ ∗∗∗” P < 0.001, “ ∗∗” P < 0.01 “ ∗” P < 0.05 , “NS” for not significant.

Field Survey

We collected noni fruits on Cousin island in the Seychelles and reared out the flies and parasitic wasps. On these noni, we found two species of Drosophila, D. sechellia and D. malerkotliana. Our survey of the Drosophila community revealed that the stage of maturity was decisive for the number and type of species that emerged from it (Fig. 2).

Figure 2 Number and species of insects emerging from Morinda citrifolia at different stages of maturity.

Each bar corresponds to one noni sample (where at least one drosophilid species emerged). The colors in the bars represent the different species of insects that emerged; the colors in the horizontal line represent the maturation stage of the noni samples that were collected. Samples 1–3 (green) corresponded to unripe, samples 4–10 (yellow) ripe and 11–17 (pink) to overripe fruits.

Only D. sechellia was found in ripe noni (stage 3), while it was also found in low numbers on some of the unripe and the overripe samples. Another Drosophila species, D. malerkotliana, was also present in unripe noni (stage 2) and was abundant in the overripe stage (stage 4) but completely absent from the ripe stage. A small sample of alternative fruit baits (banana and papaya) also yielded only these same two species of drosophilid flies.

Parasitic wasps emerged from two noni fruit, both of the overripe stage (stage 4), around seven weeks after the collection of the fruit. The morphological and molecular analysis identified the wasp species as Leptopilina boulardi. The typical developmental time of these parasitoids at 20–25 °C in the lab is approximately 4 weeks, suggesting that parasitoid oviposition occurred during the overripe stage, i.e., when D. malerkotliana was the predominant species. No wasps emerged from noni fruits that were collected during the ripe stage when D. sechellia was dominant.

Discussion

The consequences of the host shift of D. sechellia to noni have been extensively studied in the context of adaptive trait evolution and life history strategies, as it provides a good model system to gain insights into the evolution of dietary specialization and tolerance to a “toxic” resource (Jones, 1998; McBride, 2006; Matsuo et al., 2007). Inspired by evidence from phenotypic, genomic, transcriptomic and ecological studies, we hypothesized that another consequence of D. sechellia’s specialization on noni is that it provides protection from the infection by parasitoid wasps. This could cause a relaxation in the selection pressure to maintain parasitoid resistance, and thereby lead to trait loss. The evidence included the lack of immunological resistance against parasitoids wasps despite producing lamellocytes, and the changes in the sequence and expression found in genes involved in the immune response against parasitoid attack compared to closely related species. Fundamental to this hypothesis is an assumed harmful effect of noni on parasitoid wasps and a lack of successful parasitization by the local community of parasitoids of D. sechellia that feed and breed on noni in the Seychelles. Our laboratory experiment showed that a noni extract indeed significantly decreased both the number of hosts that were parasitized and the success for the parasitoid A. citri. While this species successfully parasitized and developed from both D. melanogaster and D. simulans when reared on a control medium (with yeast), a significantly higher number of the larvae on noni was not parasitized. This may have been caused by the high mortality among parasitoid wasps during prolonged exposure to noni extract during parasitization. Although these patterns are in agreement with our predictions of a parasitoid-free environment we did not have access to a strain of parasitoid wasps that is naturally occurring on the Seychelles nor did we use D. sechellia as host. A subsequent field survey in the Seychelles to explore whether D. sechellia experiences parasitization in its natural habitat on noni fruits provided further support that the shift to noni largely protects it from parasitoid attack. While a small number of parasitoids was reared from 2 of the 28 collected noni fruit, these noni fruit were overripe, at which stage they are predominantly infested by a different Drosophila species, D. malerkotliana.

It is important to mention that the policy of collection in the Seychelles is very restrictive, particularly concerning live material, which cannot be extracted from the Islands. This imposed strong limitations to the sample sizes, because all the data had to be obtained in situ. In addition, many samples were lost due to uncontrolled conditions (e.g., samples were lost due to rain washing out some of our collected fruits, and animal invasions raiding our noni samples while they were still out in the field). Despite these shortcomings, our survey confirmed that D. sechellia was the only species, and predominantly reared from the ripe stage of noni (e.g., in lightgreen to yellow fruits approx. 1 week after falling to the ground), as predicted by its tolerance to the highest concentration of toxin in this stage (Legal, Chappe & Jallon, 1994). We also recorded it at earlier maturation stages of noni, when they were unripe (less than a week after falling to the ground, and green), but they were mostly absent or rare on overripe stages (more than 2 weeks after dropping and light-yellow to white coloration). In both the unripe and the overripe stages, however, we found another species of Drosophila that we could rear from the fruits, D. malerkotliana. We also found a small number of L. boulardi parasitoids emerging from 2 of the collected noni fruits, both in the overripe stages when D. sechellia was mostly absent or very rare and D. malerkotliana was abundant. In the ripe noni fruits that we collected, from which only D. sechellia fruit flies were reared, no parasitoids were recorded. This supports our hypothesis that the specialization of D. sechellia on ripe noni fruits, containing high concentrations of octanoic acid, may provide a parasitoid-free space, which may have led to the loss of the immunological defence against parasitoids.

Parasitoids are considered an important selection force due to the heavy mortality they can inflict on other insects, thus release from this enemy could have played an important role in accelerating the specialization of D. sechellia. The mechanism by which D. sechellia would escape parasitism is, however, not clear, and at least two possibilities exist: (1) the wasps may avoid parasitizing hosts on noni at the ripe stage or (2) wasps parasitize during the ripe stage, but the parasitization is not successful as the developing parasitioids do not survive either because of the toxin of the fruit or due to some unknown resistance mechanism of the fly host. It is well known that parasitic wasps, and particularly L. boulardi can develop in D. sechellia in the laboratory (Schlenke et al., 2007; Lee et al., 2009). Unfortunately very little is known about the tritrophic system of D. sechellia feeding on noni and its parasitoids in the wild. This is likely to be due to the difficulties in collecting and exporting material from the Seychelles. Therefore we believe that our data, albeit very limited in sample sizes, is valuable to understand the natural history of this species, which is a model system for speciation and specialization adaptive evolution studies.

D. malerkotliana is considered an invasive species, which has been found to be a strong competitor of D. sechellia (Lachaise et al., 2008). A study showed that this species (together with other species from the ananassae group) are able to encapsulate and kill parasitoid eggs without melanization by means of gigantic cells with filamentous projections and multiple nuclei (named multinucleated giant hemocyte, MGH). These cells share some properties with lamellocytes, such as the ability to encapsulate foreign objects, but differ considerably in their morphology and function (Márkus et al., 2015). Potentially D. malerkotliana’s ability to resist parasitoid wasps has helped it to invade the decaying noni fruits, where Leptopilina parasitoids have been found, thus filling a niche in exploiting rotting stages of the noni. On other islands in the Seychelles archipelago, several other drosophilid species have also been recorded (Lachaise et al., 2008), but our small survey (including alternative fruit baits) did not yield any other Drosophila species on Cousin. Cousin has a special conservation status, and all non-native plants are systematically removed from the island, which severely restricts the alternative fruit resources for breeding for Drosophila and its parasitoids.

Conclusion

Collectively, our data provides support for the hypothesis that a consequence of D. sechellia’ s ecological shift to noni may have protected this species from the parasitic wasps. Although the sample sizes are small and do not provide conclusive evidence on their own, they are in line with our predictions. Ideally, we would need to test a population of L. boulardi from the Seychelles to assess whether the toxins in noni are indeed harmful to them, and for their ability to parasitize D. sechellia that are breeding on ripe noni fruits. This could reveal whether the host shift to noni has indeed provided D. sechellia with an enemy-free space, which could explain the secondary loss of an immune response to defend themselves against parasitoids.

Supplemental Information

Supplemental Information 1 Raw data obtained from lab experiment containing counts of flies (Drosophila melanogaster and Drosophila simulans) and wasps (Asobara citri), and used for Fig. 1

Click here for additional data file.

Supplemental Information 2 Raw data obtained from field survey containing counts of flies (Drosophila sechellia and Drosophila malerkotliana) and wasps (Leptopilina boulardi) for each noni sample, and used for Fig. 2

Click here for additional data file.

We would like to thank Marie-Louise Cariou for the taxonomic identification of the Drosophila species and Fabrice Vavre (Laboratory of Biometry and Evolutionary Biology, CNRS-Lyon) for conducting the amplification of the COI fragment of the parasitic wasps. To Jan Komdeur and Sjouke Kingma for information and contacts in Cousin. To the NGO Nature Seychelles and the team of volunteers on Cousin island for supporting the field work, in particular to Alex Underwood for assisting with the collection sites. To Jesse Jorna and Ralph Bennen for help with the lab experiment. We would like to also thank the anonymous reviewers for the constructive comments, which helped improve the manuscript.

Additional Information and Declarations

Competing Interests

Author Contributions

Field Study Permissions

Data Availability

Laura Salazar-Jaramillo is researcher at the Vidarium-Nutrition, Health and Wellness Research Center. The authors declare there are no competing interests.

Laura Salazar-Jaramillo conceived and designed the experiments, performed the experiments, analyzed the data, prepared figures and/or tables, authored or reviewed drafts of the paper, and approved the final draft.

Bregje Wertheim conceived and designed the experiments, analyzed the data, authored or reviewed drafts of the paper, and approved the final draft.

The following information was supplied relating to field study approvals (i.e., approving body and any reference numbers):

Field experiments and material collection was approved by the Seychelles Bureau of Standards (Ref A0157).

The following information was supplied regarding data availability:

Field data and scripts are available at GitHub: https://github.com/lauraalazar/Dsechellia-parasitoid.

The discussed sequences are publicly available at the ENA: PRJEB15540.

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
