# Peer review of "Does Drosophila sechellia escape parasitoid attack by feeding on a toxic resource?"

_PeerJ, doi:10.7717/peerj.10528_

## Round 0.1 · original submission · Major Revisions

Now reviewers have revised the manuscript. The reviewers think the manuscript is sound, and that findings are interesting. However, both reviewers agree that the manuscript needs much more work and substantial improvement. I greatly agree with them.

The introduction needs further work, especially in terms of the hypothesis. Pay particular attention to the small sample size used, which is critical to support conclusions. Notice that for both reviewers, this is a key aspect and needs to be addressed in the rebuttal letter. I remind authors that PeerJ evaluates articles based only on an objective determination of scientific and methodological soundness, not on subjective determinations of 'impact,' 'novelty' or 'interest'.

Authors need to provide raw data. There are comments regarding typos, caption side in figures and all other observations regarding the format which diminishes the quality standard needed for this journal. That is basic. I strongly encourage the authors to take all the reviewers’ suggestions and effectively include them in the text, and if not, clearly justify why you don't. Please, provide a fully detailed point by point letter. I highlight to the authors that the next step is a critical stage for a positive decision regarding the acceptance of this work.

Reviewer 1 ·

Basic reporting

The language needs some revision, as pointed out in my specific comments. The scholarship needs substantial additions, as I have pointed out in my specific comments. The structure of the Introduction needs to be expanded, and the structure of the Methods and Results don’t align. The figures need clearer captions in places. The authors did supply the raw data.

Experimental design

This is an observational study. The research question is defined, but hypothesis lacks rationale and scholarly context. How the work contributes to field is not completely clear because of lack of scholarship and reporting of what is already known. The design of the observational study is appropriately rigorous. However, the description of methods needs some revision, as I have stated in some specific comments.

Validity of the findings

The study is valid, but replication is limited. Consequently, the small sample sizes make this study questionable as a stand-alone paper. I suggest adding two components: 1) the unpublished results described in the Discussion section (lines 187-189) and/or 2) an expansion of the hypothesis posed here. The latter is critical for revising the manuscript. By truly explaining the hypothesis (or alternative hypotheses), this could be more of an idea paper inspired by previous experimental results combined with the modest yet important observational data reported here already. The scientific soundness is fine, but the contribution is modest.

Additional comments

This study provides rare observations about the natural history and tritrophic ecology of Drosophila seychellia, which has attracted interest among evolutionary biologists and geneticists because of its exception ecological specialization. The main field observation reported here, that D. seychellia was not found to be parasitized while a congener using the same fruit at different stages was found to be parasitized, is quite interesting even if based on a small sample of observations. While interesting, the small sample size limits the contribution of this study and manuscript. I suggest bolstering these field observations with either an expansion of the conceptual framework beyond what is typical for a data paper or combining these observations with additional empirical results. Some interesting unpublished results cited in the Discussion section would seem to be a good choice.

In its current form, the manuscript suffers from various problems in presentation of ideas, methods, results, and discussion. These issues need to be addressed before the manuscript can be considered for publication. I have provided many suggestions for revisions in my specific comments below.

Specific comments
Lines 12 and 38. Replace “specialized” with “specializes.”

Lines 27-28. This is a matter of style, so not necessary. However, I suggest replacing “speciation and to be a key factor underlying the diversity of insects Ehrlich and Raven (1964)…” with “evolutionary diversification of insects (Ehrlich and Raven 1964; …).”

Lines 33-34. The sentence starting with “One of the pioneer species…” is not clear to me. Rephrase to explain what you mean.

Line 47. Remove the comma. It is not grammatically correct.

Lines 47-49. Replace “provides this species with a reproductive advantage by being able to access the food source during an earlier time in the fruit’s development” with “provides this species with a unique ecological niche among coexisting and potentially competing Drosophila species.” I think this is the important point, and the details of the niche could be added to this point.

Line 50. Add an apostrophe (D. sechellia’s).

Line 52. Cite addition references here.

Lines 57-58. What is the rationale for this hypothesis? There are many species of insects that feed on toxic foods and still experience substantial mortality from parasitoids. I suggest you consider how host-specific the parasitoids are. In fact, the host range of the parasitoid observed in this study has been studied by Schlenke and coauthors, so there is relevant information to cite. According to a meta-analysis by Zvereva and Koslov 2016, Ecological Monographs 86: 107-124, chemical defenses of herbivorous insects are generally effective against generalist predators but not parasitoids and specialist predators. Generalist parasitoids show a wide range of responses, so it is plausible to hypothesize that toxic hosts might be defended against generalist parasitoids.

Line 59. In this paragraph, it is worth discussing related work in which fruit flies gain resistance against parasitoids from their potentially toxic diet (Milan et al. 2012, Current Biology 22: 488). I suggest summarizing this study because it involves the parasitoid species found here and gives a precedent for your hypothesis. There is also work on geographic variation in anti-parasitoid resistance in Drosophila by Kraaijeveld and colleagues (late 1990’s and early 2000’s). There is also relevant scholarship on ecological immunology that mostly deals with insect herbivores outside of Drosophila (e.g., reviewed by Singer et al. 2014, Integrative and Comparative Biology 54: 913).

Line 75. Provide more details about these other cases of Drosophila lacking melanotic encapsulation ability. What are their ecological characteristics? Do they feed on toxic substrates? This is all relevant context for this study.

Line 102. Elaborate on your hypothesis providing your rationale. This should probably go earlier, when the hypothesis is first described.

Line 112. First describe the study system. What is already known about the natural history? Or would this be the first study to look at the natural history of tritrophic interactions in this system?

Line 116. Replace “of” with “off.” To correct the grammar, replace “A total of “ with “In total,” at the beginning of the next sentence.

Line 118. On what basis were these categories determined? Did they look or smell different over these time periods? As described, the classification seems arbitrary.

Line 120. Be more specific. The time of exposure could be an important factor determining parasitoid access.

Line 125. For ecological clarity, I suggest referring to this part of the study as a survey of fruit-feeding Drosophila in the same community. Later you could refer to it as the community survey.

Line 127. Replace “were” with “was” for proper grammar.
Lines 131-132. Justify the exclusive use of morphological characters for the fruit fleis when you used DNA barcoding on the parasitoids.

Line 137. Replace “characterization” with “community survey.”

Line 140. Where are the results from the other fruit baits? Only the noni fruit data are shown in Table 1 and described in the Results section. If the passage in the Discussion mentioning the loss of data due to animals and weather caused there to be no data from the broader fruit bait survey, then don’t describe this survey in the methods of this paper. As it stands, the methods create the expectation that all of the results from them will be reported.

Figure 1 caption. Label all the panels with letters to you can refer to them unambiguously. “Column” is not clear enough. “Circles” is misspelled. Give the species name of the wasp. Last sentence: replace “is” with “are.”

Figure 2 legend and caption. If “alcohol” could be replaced with “ethanol” it would be clearer.

Line 150. I suggest adding “ecological” to the list of types of studies.

Line 156. This paragraph should be expanded so you can explain your hypothesis and how these results relate to it in greater detail.

Line 174. Describe these studies further. It’s important to know the circumstances under which L. boulardi can develop in D. sechellia. Did the flies feed on noni fruit or artificial diet in this experiment? If the former, the present results suggest that the parasitoid avoids ovipositing in flies eating ripe fruit. If the latter, resistance might occur either through reduced parasitoid attack or reduced parasitoid survival in toxic hosts.

Line 180. The topic (first) sentence here suggests the paragraph will be about parasitoids promoting evolutionary diversification of their hosts. Either change the first sentence or make the paragraph about that topic. The existing content of the paragraph should probably be moved to the previous paragraph following my comment there.

Lines 187-189. This is very interesting! Why isn’t this experiment/result included formally in this paper?

Line 193. Remove the comma.

Line 200. How so? This conclusion was never explained.

Line 209. There is a typo: it should be “conducting.”

References. Give the full reference for Andrade Lopez (2015). I gather this is a published thesis, so there should be a university and it should be identified as a thesis. The journal title in Farine et al. has a typo.

Reviewer 2 ·

Basic reporting

In general, the study is fairly clearly presented. There are a fair number of instances of awkward language and some errors, but this would not be too difficult to clean up with a close reading of the manuscript with attention to my comments below (general comments).

Overall structure is fine and the background sufficient. The overview of previous work examining D. sechellia’s lack of parasitoid resistance, both experimental and through comparative genomics was very interesting and well presented.

Figure 1 and 2 seem unnecessary. All the data in figure 1 was previouly published and explained in the introduction. It does not seem necessary to reproduce it here as a figure.
See comments below regarding Figure 2.

Experimental design

I appreciate the limitations the authors had, but 28 noni fruit seems a pretty small sample size for this sort of study, and just 9 wasps from two rotting fruit (a stage that hosted both Drosophila species) is not strong evidence of a pattern.

It seems possible that wasps don’t parasitize larvae of either Drosophila species until the fruit is rotting due to use of odors in host location and perhaps physical barriers (e.g., fruits are too hard). Still this would be consistent with the idea that using ripe but not rotting fruit provides some enemy free space.

It is not entirely clear what the artificial baits were for. Just to see what might be attracted? To compare attractiveness of the baits/substrate use? Are the only results from this that, yes, the two species of Drosophila were found in the baits? Was there a difference in attraction to different baits between the species? Were D. sechellia attracted to the papaya or banana? Was the other species attracted to the noni fruit bait?

It would have been more revealing to allow the Drosophila in the baits to be exposed to parasitoids to assess parasitism rates. Even better would have been to present wasps with host larvae of each Drosophila species and allow those larvae to develop on various substrates and assess parasitism rates. This would be key in assessing if the fruit toxins actually harm the wasps or if they are simply attracted to rotting fruit due to the volatiles or whatever

Validity of the findings

The authors conclude from a limited sampling study that because they reared parasitoids from only rotting fruits that are less frequently fed on by D. sechellia, that feeding on ripe nona has relaxed parasitoid pressure and resulted in diminished physiological defenses. This is logical, and I tend to think the author’s story is likely, but it is a bit of a leap based on the limited evidence they present. Just 9 wasps from two rotting fruit (a stage that hosted both Drosophila species) is not strong evidence.

“D. malerkotliana, was abundant in unripe noni and also present in the rotting stage but completely absent from the ripe stage.” – But this is just 4 fruit!

If typical development time of the wasp is four weeks, why did they emerge after seven weeks?

Are the noni toxins toxic to the parasitoid wasps?

Are the wasps attracted to ripe noni fruit?

Are wasps able to penetrate the ripe fruits?

Why were no D. malerkotliana in unripe fruits parasitized if toxin levels are low?
I find the hypothetical curves in Figure 2 very unconvincing. What is the basis of these curves? Show the data. Also, the pie charts are misleading because the low number of experimental units = fruit is not apparent.

If the author did a study and found that noni extract is harmful to adult parasitoid wasps, why isn’t that study described and include in this paper instead of referring to unpublished results? Still, those conditions appear artificial and did not involve D. sechellia.

Additional comments

p. 1
while closely-related species (delete “the”, this article (“the”) is used many times in the manuscript in places where it is not necessary and sounds awkward)
L. 26 I would say parasitic organisms
L. 28 refs should be in ()
Radiation followed by host shifts? Do you mean the opposite? And it’s unclear how the listed factors facilitate radiation. (radiation misspelled, enemy free space)
What is meant by “traces left in the genome”?

The authors say that [genome] sequencing of Drosophila has allowed comparative analyses but they don’t mention any such analyses or the insights they have provided.

…a small proportion of adults also found in other substrates(?) Adults aren’t found in substrates. Same applies to D. simulans.

Adults on substrates?

P. 2

L. 55 driving trait loss

L. 56 apply to

L. 57 with protection from attack

L. 59 significant mortality factor?

L. 61 In this process, the parasitoid egg...

L. 63 it should be stated that the hemocytes become melanized

L. 72 infection of which wasp species?

L. 80 associated with

P. 3

L.104 Cousin or Couisin?

L. 113 drosophilid

We collected noni fruit that had fallen from [or off] the plant,

L. 131 Drosophilidae is not italicized.

P. 4

Table 1 names need a space, e.g., D.malerkotliana

Figure 1. These results were previously published, I do not think they need to be placed in a figure such as this here. That said, check the legend for spelling/grammatical errors.

Figure 2 is also not necessary, and is actually misleading. The hypothetical curves are just that, hypothetical. If they are based on data, show the data. The pie charts are total numbers of insects, but this is misleading because fruit sample sizes were much lower.

P. 7

L. 152 This could cause a relaxation…

L. 157 feed and breed

---

## Round 0.2 · Minor Revisions

We now have the reviewers' comments. This was a good job. However, new changes are suggested that are necessary to refine and / or improve some aspects of the manuscript. The inclusion of these aspects or their refutation, should be detailed point by point in a letter to the editor, together with a second version of the manuscript. Raw data must be included.

Reviewer 1 ·

Basic reporting

Generally good. The language still needs some minor revision, as pointed out in my specific comments. One of the figures needs some revision. I didn’t see raw data supplied by the authors. Did I miss it?

Experimental design

The research question is defined, and the hypothesis is now described with much more scholarly context (perhaps a bit more than needed, but it’s fine). The design of the observational study is appropriately rigorous. The design of the newly added lab experiment is sound. However, the description of statistical methods still needs some revision, as I have stated in some specific comments.

Validity of the findings

The study is valid, and the additional scholarship and lab experiment were helpful in validating it further. The overall weight of the evidence from this study and others make a more compelling case than could be easily understood from the previous version of the manuscript.

Additional comments

This study provides rare and valuable observations about the natural history and tritrophic ecology of Drosophila seychellia, which has attracted interest among evolutionary biologists and geneticists because of its exceptional ecological specialization. The main field observation reported here, that D. seychellia was not parasitized while a congener using the same fruit at different stages was parasitized, is quite interesting even if based on a small sample of observations. I am pleased to see that this observational study is bolstered with more discussion of the previous scholarship and a laboratory study that was previously only mentioned in passing. The evidence suggests the ripe noni fruit deters parasitoids because it is toxic to them (according to the lab study).

The revisions in this version were very helpful. A minor revision based on the comments below should make this manuscript ready for publication.

Specific comments

Line 25. Delete “drastically” because there isn’t enough evidence to support it.
Lines 33-34 (and perhaps elsewhere). Follow a conventional order when listing in-text citations (usually by year from oldest to newest).
Line 35. “Affects mating in pea aphids”
Line 42. “Has been described”
Line 44. “Dietary specialist”
Line 50. Cite reference to support this statement.
Line 53 and in bibliography. Lanno et al. 2018
Line 59. Remove the comma between “toxicity” and “provides.”
Line 61. Add a comma between “species” and “we.”
Line 71. Each is a singular subject, so the verb should be “is” instead of “are.”
Line 75. “Herbivorous insects”
Line 85. Remove the comma.
Line 87. Remove the quotation marks around “foreign body.”
Line 123. I don’t see the need to use the metaphor of “machinery” when you can just say that they have the genes, but they no longer function properly due to limited expression in response to parasitoid attack.
Lines 127-129. I suggest making the distinction here between direct toxicity of noni fruit and toxicity of fly larvae which have fed on noni fruit. Both are possible at this point, and later evidence is presented for the former possibility.
Line 154. UC San Diego (not San Diego University)
Line 163. The sample size of 4 replicates is important. It is not clear to me if this is the number of experimental units treated as independent replicates in the statistical analysis. I would like to see an explicit description in the methods.
Line 170. Move parenthesis: Salazar-Jaramillo et al. (2014)
Line 188. Don’t capitalize drosophilid.
Line 194. Specify the type of alcohol (ethanol?).
Line 197. Give the institution for Prof. Cariou.
Lines 204-207. The repetition of the hypothesis and methods is not needed. Delete this paragraph.
Lines 212-213. These P-values seem pretty small for a sample size of 4 (see comment above), so I’m wondering about the unit of replication in the statistical analysis versus the experiment.
Line 217. The wording here could be clearer: “left unparasitized.” I think you mean that fewer hosts were attacked by parasitoids in the noni medium.
Lines 226-227. In this case, it would be clearer to say “Our survey of the Drosophila community” because you just mentioned flies and wasps.
Line 230. “Was also present”
Line 233. Don’t capitalize or italicize drosophilid.
Line 237. Clearer to say “parasitoid oviposition occurred during the overripe stage.”
Fig. 1 B. The title has “parasitization” misspelled.
Fig. 1 caption. Describe the figure so the reader can interpret it without repeating the methods. Your Fig. 2 caption is a good example of what to do.
Fig. 1 and caption. The graphical depiction of data suggests a contingency test was used, whereas the methods say a GLM was used. The figure should reflect the comparison made in the analysis, and thus should have error bars in panel B.
Line 243. “Evolution of dietary specialization”
Line 248. “Immunological resistance against parasitoids”
Line 263. “Infested” not “invested.”
Lines 264-267. Don’t repeat the methods here. You can just say that the legal restrictions limited the research to an observational study with limited sample sizes.

---

## Round 0.3 · accepted · Accept

The new version now includes all the comments suggested and the manuscript has improved quite a lot. Raw data are provided. Congratulations to the authors for this !!